# Bunyaviruses Affect Growth, Sporulation, and Elicitin Production in *Phytophthora cactorum*

**DOI:** 10.3390/v14122596

**Published:** 2022-11-22

**Authors:** Anna Poimala, Milica Raco, Tuuli Haikonen, Martin Černý, Päivi Parikka, Jarkko Hantula, Eeva J. Vainio

**Affiliations:** 1Natural Resources Institute Finland (Luke), Latokartanonkaari 9, FI-00790 Helsinki, Finland; 2Phytophthora Research Centre, Department of Forest Protection and Wildlife Management, Faculty of Forestry and Wood Technology, Mendel University in Brno, Zemědělská 3, 613 00 Brno, Czech Republic; 3Natural Resources Institute Finland, Toivonlinnantie 518, FI-21500 Piikkiö, Finland; 4Phytophthora Research Centre, Department of Molecular Biology and Radiobiology, Faculty of AgriSciences, Mendel University in Brno, Zemědělská 1, 613 00 Brno, Czech Republic; 5Natural Resources Institute Finland, Humppilantie 18, FI-31600 Jokioinen, Finland

**Keywords:** mycovirus, *Bunyaviridae*, oomycetes, *Phytophthora cactorum*, virus curing, PEG 8000

## Abstract

*Phytophthora cactorum* is an important oomycetous plant pathogen with numerous host plant species, including garden strawberry (*Fragaria* × *ananassa*) and silver birch (*Betula pendula*). *P. cactorum* also hosts mycoviruses, but their phenotypic effects on the host oomycete have not been studied earlier. In the present study, we tested polyethylene glycol (PEG)-induced water stress for virus curing and created an isogenic virus-free isolate for testing viral effects in pair with the original isolate. Phytophthora cactorum bunya-like viruses 1 and 2 (PcBV1 & 2) significantly reduced hyphal growth of the *P. cactorum* host isolate, as well as sporangia production and size. Transcriptomic and proteomic analyses revealed an increase in the production of elicitins due to bunyavirus infection. However, the presence of bunyaviruses did not seem to alter the pathogenicity of *P. cactorum*. Virus transmission through anastomosis was unsuccessful in vitro.

## 1. Introduction

The oomycete genus *Phytophthora* includes notorious plant pathogens that severely threaten agricultural crops and forest trees worldwide. *Phytophthora cactorum* is an omnivorous pathogen that can infect a wide variety of plant species [1]. On cultivated strawberry, it is the causative agent of crown rot as well as leather rot of fruits. Both diseases cause economic losses in strawberry production globally [1,2]. The degree of resistance to the disease varies among strawberry cultivars [3]. Crown and leather rots are controlled mainly by propagating plants for planting from healthy stocks and by planting strawberries on ridges to improve drainage and aeration (hilling), but soil sanitation and fungicides are also used [4].

*Phytophthora cactorum* is a genetically variable species with two cosmopolitan lineages associated with strawberry and woody species (apple, oak, ornamental trees), respectively [5]. Finnish isolates from birch have been inferred to represent the most early-diverging lineage within the species and separate phylogenetically from the two cosmopolitan lineages [5]. Isolates from woody species, including Finnish isolates from birch, have caused little or no crown rot symptoms when inoculated in strawberry, but strawberry isolates have caused necrotic lesions on *B. pendula* [6,7,8]. *Phytophthora cactorum* can persist in the soil or symptomless plants for planting as sexual oospores or chlamydospores that germinate under wet conditions to produce mycelium and sporangia, which release motile zoospores to infect plant crowns or fruits [9].

Fungi and oomycetes host viruses that cause persistent, usually asymptomatic, infections in their host. Viruses reported from *Phytophthora* species include many alphaendornaviruses infecting *P. cactorum, Phytophthora ramorum*, a yet unnamed species designated as *Phytophthora* taxon douglas fir, and a *Phytophthora* sp. from asparagus [10,11,12]. Even single host species may be inhabited by several bunyaviruses, as *P. cactorum* and *Phytophthora condilina* have been found to host three and 13 putative species, respectively [10,13]. A virus with affinities to *Totiviridae* has been found to be the most common virus in *P. cactorum* [10,14], and two related viruses have also been found infecting *P. condilina* [13]. A member of the proposed genus “Ustivirus” has also been reported from a single isolate of *P. cactorum* [10]. Viruses infecting the potato late blight pathogen *Phytophthora infestans* include viruses resembling *Narnaviridae* [15] and the proposed family “Fusagraviridae” [16], as well as two novel unclassified viruses [17,18]. A novel virus with a resemblance to members of *Ghabrivirales* was also recently reported from *Phytophthora pluvialis* [19]. In addition, multiple novel viruses were recorded recently in *Phytophthora castaneae,* including viruses with phylogenetic affinities to families *Endornaviridae, Narnaviridae, Totiviridae, Megabirnaviridae,* and “Fusagraviridae” in addition to members of *Bunyavirales* [20]. A giant virus has also been found integrated in the *Phytophthora parasitica* genome [21]. In the first population-level study on *Phytophthora* viruses [10], virus prevalence in a *P. cactorum* population infecting garden strawberry was found to be very high. The most common viruses were toti-like and endornaviruses occurring in 94% and 57% of isolates, respectively, whereas bunya-like and usti-like viruses occurred much more rarely (4.5% and 1.1% of isolates, respectively). Viral populations infecting isolates from birch and strawberry were also suggested to be different [10].

Although mycoviral infections are most often latent, some mycoviruses can cause phenotypic debilitation in their plant-pathogenic host fungi, and those have been widely studied in the search of potential biocontrol agents [22,23,24,25]. The most successful virocontrol application has been the artificial introduction of hypovirulence-causing Cryphonectria hypovirus 1 (CHV1) to the host fungus *Cryphonectria parasitica* to control Chestnut blight disease in Europe [26]. The effects of mycoviruses can also be seen as alterations in gene expression. For example, host debilitation inducing Heterobasidion partitivirus 13 and hypovirulence causing Rosellinia necatrix megabirnavirus 1 significantly affected the transcription of 683 and 1160 genes in their hosts, respectively [23,27]. Transcriptome analyses of *Cryphonectria* hypoviruses, *Fusarium graminearum,* and *Botryosphaeria dothidea* viruses as well as Sclerotinia sclerotiorum debilitation-associated RNA virus revealed that the expression levels of specific genes varied between virus-free and virus-infected isolates, but also between phylogenetically different viruses as well as among virus and host strains [28,29,30,31,32]. Despite the importance of oomycete pathogens, very little is known about viral effects on these hosts. However, coinfection of two endornaviruses in *Phytophthora* sp. from asparagus has been shown to enhance sporangia production in at least some host isolates, inhibit mycelial growth, and potentially modify fungicide sensitivities of the host [11]. Phytophthora infestans RNA virus 2 (PiRV-2), a novel unclassified virus, has also been shown to stimulate sporangia production in *P. infestans* [33]. It also induced the upregulation of the host ribosomal and histone protein genes, as well as genes for flagellum-related proteins, proteins with epidermal growth factor (EGF)-like conserved site, and genes involved in the glycolytic process in *P. infestans* [33].

The order *Bunyavirales* includes viruses with linear, single-stranded, negative-sense, or ambisense RNA genomes classified into 14 families, four subfamilies, 60 genera, and 496 species [34]. Their genomes generally comprise three unique molecules designated L (large), M (medium), and S (small), with sizes of 11–19 kb. The large segment generally codes for an RNA-dependent RNA polymerase (RdRP), whereas the M segment includes genes for two external glycoproteins, and the S-segment encodes a putative nucleocapsid and silencing suppressors [35]. The diversity of bunyaviruses has been increasing due to metagenomic studies [36], and currently, classified species of *Bunyavirales* are found in plants, invertebrate, and vertebrate hosts. They are also transmitted by arthropod and mammalian vectors [37]. Viruses with genomic similarities to bunyaviruses have more recently also been described from fungi and oomycetes. Fungal species hosting these unclassified bunya-like viruses include several potential phytopathogens [38,39,40,41,42] as well as plant endophytes [43]. Further hosts include the shiitake mushroom (*Lentinula edodes*) [44] and a marine fungus, *Penicillium roseopurpureum* [45]. In recent years, bunyaviruses have also been detected in *Oomycetes*. Pythium polare bunya-like RNA virus 1 (PpBRV1) was reported infecting *Globisporangium* (syn. *Pythium*) *polare* [46], and eight novel (−)ssRNA viruses resembling members of *Bunyavirales* were recently described in *Halophytophthora* [47]. Bunya-like viruses were also reported from *Plasmopara viticola* lesions in grapevine [48]. Furthermore, 13 bunya-like viruses have been detected in *P. condilina* [13], three in *P. cactorum* [10], and one in *P. castaneae* [20]. No direct effects of bunyavirus infection on the growth or virulence of their fungal or oomycete hosts have been recorded. However, Wang et al. [49] detected Macrophomina phaseolina mycobunyaviruses 1–4 (MpMBV1–4) only in hypovirulent fungal isolates.

Fungal viruses usually do not have extracellular infective particles, but they replicate in the cytoplasm of their hosts and transmit intracellularly during cell division and hyphal anastomosis (horizontal transmission) as well as by sexual and asexual spores (vertical transmission) [34,50]. They are usually readily transmitted in vitro between compatible and even incompatible fungal isolates in dual cultures. Successful transmission of an oomycete virus, the unclassified Phytophthora infestans RNA virus 2 (PiRV-2), through a hyphal anastomosis between isogenic hosts has also been reported by Cai et al. [33]. However, the virus did not transfer between different host genotypes, indicating possible hyphal incompatibility between isolates. Nevertheless, PiRV-2 was transmitted to individual zoospores with 100% frequency [34]. In the present study, we aimed to investigate virus effects on *P. cactorum* to identify oomycetal viruses with biocontrol potential. For this purpose, we tested methods for curing and transferring *Phytophthora* viruses. The effects of bunyaviruses were examined by comparing the successfully obtained isogenic virus-infected and virus-free isolates that were investigated for their growth rate, sporulation, gene expression, protein abundance, and pathogenicity on two host species (strawberry and birch).

## 2. Materials and Methods

### 2.1. Phytophthora Isolates

*P. cactorum* isolates used in this study were obtained from the isolate collection of Natural Resources Institute Finland (Table 1). They had been isolated from strawberry plants showing crown rot symptoms in three farm locations. The viruses hosted by these isolates have been reported previously [10].

### 2.2. Curing Virus Infections

Water availability-related matrix stress was induced by a series of different polyethylene glycol (PEG) 8000 (VWR LifeScience) concentrations added to potato dextrose broth (PDB; Difco^TM^) growth medium (modified from [51]). Water potential gradients ranging from -1 MPa to -15 MPa were created. The amount of PEG 8000 in a gram of cultivation broth was calculated based on the Michel equation: Ψ (water potential) = 1.29 [PEG]2T–140[PEG]2–4 [PEG], and the value was adjusted to the culture temperature of 20 °C [52]. An agar plug with actively growing mycelium was placed in 50 mL of autoclaved PDB media containing a targeted concentration of PEG 8000. Three isolates (PhF9, PhF66, PhF79; Table 1) hosting endornaviruses (+ssRNA), toti-like viruses (dsRNA), bunyaviruses (-ssRNA), and an usti-like virus (dsRNA) in two replicates and water potentials ranging from −1 MPa to −15 MPa were included in the test. After three weeks, pieces of newly growing mycelia from the liquid media were transferred to potato dextrose agar (PDA; Difco^TM^). After one week, the isolates were transferred on modified orange serum (MOS) agar (HiMedia Laboratories) covered with cellophane membrane. After two weeks of growth, mycelia were harvested for RNA extraction. RNA was extracted and reverse transcribed according to Poimala et al. [10], and the resulting cDNAs were screened with PCR for the presence of viruses using specific primers [10]. The disappearance of the viruses was further confirmed by RNA seq, where RNA was extracted with a Spectrum plant total RNA kit (Sigma–Aldrich, Saint Louis, MO, USA) and sent to a sequencing facility (Macrogen, Seoul, Rep. of Korea) for library preparation and RNA-seq, as previously reported [10]. The isolate cured from viruses was designated asPhF66–.

The presence and absence of the bunyaviruses in PhF66 and PhF66–, respectively, was also re-checked before morphological, sporulation, and proteomic analyses. Here, RNAzol (Sigma-Aldrich) was used for RNA extraction and a High-capacity cDNA kit (Thermo Fisher) for reverse transcription, as in Raco et al. [20].

### 2.3. Virus Transfer Experiment

Isolates PhF38, PhF66, PhF79, and PhF101 (Table 1) were used as donor strains and plated as dual cultures with the virus-free recipient PhF66– with six replicates on malt agar plates to enable the formation of hyphal contacts and subsequent viral transmission. Mycelial samples were taken from the recipient side of the culture after 21 and 64 days and analyzed for the presence of the corresponding viruses using RNA extraction and RT-PCR with virus-specific primers using the original virus-containing PhF66 as a positive control [10].

### 2.4. Growth Tests

The phenotypes of PhF66 and PhF66– were compared by testing their growth rates on 2% MEA plates at 20 °C. The tests were conducted using 22 replicates of each isolate. Each isolate was inoculated in the center of the plate as a mycelial plug with a diameter of 5 mm. Mycelial areas were marked on the plates two and six days after growth initiation, after which the plates were scanned. The surface area covered by the fungal mycelium was determined from the images using ImageJ program. Radial growth rates were calculated in Excel. Furthermore, 12 apple fruits (‘Golden Delicious’) per isolate were inoculated by drilling a 10 mm hole to the surface with a Ø 10 mm cork borer and inserting a mycelial plug containing mycelia (Ø 5 mm) from a 13-day-old culture to the bottom. After six days at +22 °C, the areas of the brown lesions emerging on the apple surfaces were measured. The differences in growth between the isolates were determined by one-way ANOVA (growths on MEA) or independent samples Mann Whitney U-test (lesions in apple) in IBM SPSS Statistics v27.

### 2.5. Sporulation Tests

The isolates PhF66 and PhF66– were grown on clarified 11 mL vegetable juice agar [VA; 100 mL/L vegetable juice DM Bio Gemüse Saft (DM-Drogerie Markt, Karlsruhe, Germany), 3 g/L of CaCO_3_, 18 g/l of agar (VWR LifeScience)] in 90 mm diameter Petri dishes at 20 °C in the dark. To obtain zoospores, three experiments were conducted. In experiment 1, three replicates per isolate were used. For each of them, a mycelial plug containing mycelia (Ø 5 mm) was placed in the center of a Petri dish containing 11 mL clarified VA. After four and five weeks, cultures were flooded with 12 mL sterile demineralized water for 24 to 48 h under daylight at 20 °C. Water was decanted and replaced two to three times per day. For experiments 2 and 3, two 15 × 15 mm agar plugs from three-, five-, and six-day-old cultures (four plates per isolate; two plugs per plate) were cut from the same area for each replicate (one cut closer to the edge of a Petri dish, and the other closer to the agar plug) were placed in an empty 90 mm diameter Petri dish and flooded with 12 mL of sterile demineralized water. In experiment 2, the water was replaced after 12 h, 16 h, 20 h, and 22 h with 10% pond water (100 mL/L non-sterile pond water, 900 mL distilled autoclaved water). In experiment 3, the water was replaced with sterile demineralized water at 3.5 h, with Volvic^®^ mineral water (Danone, Clermont-Ferrand, France) at 7 h and 21 h, and with Volvic^®^ mineral water amended with approximately 1 mL of 2 days old non-sterile soil extract (250 mL of soil taken under asymptomatic *Juglans regia* L. trees flooded with 2 L of water) at 24 h. After the incubation period under daylight at 20 °C, zoospore formation and release were stimulated by cold shock (60 min in +4 °C). Zoospore suspension was prepared, and spore counts were determined, as in Harris & Webber [53]. A longer incubation of 120 min at 7 °C prior to the 75 min incubation at 20 °C was also tested for stimulating zoospore release. The agar plates (experiment 1), and agar plugs (experiments 2 and 3) were inspected under a microscope to follow sporangia formation before each change of water. Zoospore counts were determined in a Bürker counting chamber. To evaluate sporulation levels, four 0.5 cm × 0.5 cm squares were cut from the four compass points on a Petri dish with five-day-old mycelia from the area of 0.5 cm to 1.0 cm from the center of the agar plug. Sporangia formation was stimulated as in the experiment number 3. The number of sporangia produced by PhF66 and PhF66– was estimated by counting all sporangia, including freshly forming, already formed (mature), and open sporangia in these four agar squares on one plate per isolate. Statistical differences were obtained by two sample *t*-test in R (v 3.6.3).

### 2.6. Morphology

The dimensions of 50 randomly selected sporangia and 50 gametangia (oogonia and antheridia) were measured from the isolates PhF66 and PhF66–. Microscopic examinations and measurements were performed as in Jung et al. [54,55], except that the sporangia for measurements were produced as in the sporulation experiment 3. Characteristics of mature gametangia (antheridia and oogonia) were examined on clarified VA after 28–35 days at 20 °C in the dark. Abortion rates were estimated from three plates per isolate by counting 100 oogonia in two 15 × 15 mm squares, one cut from the area closer to the edge, and the other from the area closer to the center of the plate. Microscopic measurements of all structures at ×400 were made using an optical microscope (Motic© BA410E), camera (Moticam 5 + 5.0 MP), and camera software (Motic© Images Plus 3.0). The photographs were taken using a compound microscope (Zeiss Axioimager.Z2, Carl Zeiss AG, Oberkochen, Germany), a digital camera (Zeiss Axiocam ICc5), and biometric software (Zeiss ZEN). Statistical differences were obtained by Bonferroni corrected two sample *t*-test in R (v 3.6.3).

### 2.7. RNA Extraction and RNA-Seq

Prior to total RNA extraction, the isolates were grown on MOS agar plates on cellophane membranes. The mycelia were harvested after two weeks, freeze-dried, and ground in liquid nitrogen, and total RNA was extracted using Spectrum Plant total RNA kit (Merck) from three technical replicates of PhF66 and PhF66– (i.e., mycelia originating from one plate). These six samples were considered biological replicate 1. Furthermore, RNA from two more biological replicates of the isogenic strains (in two technical repeats) were extracted separately in the same way, resulting in four additional RNA-seq samples. The RNA of each sample was quantified with a NanoDrop 2000 Spectrophotometer (Thermo scientific, Waltham, MA, USA). The samples were sent to a sequencing facility (Novogene UK Company Ltd., London, UK), where RNA integrity was assessed, eukaryotic strand-specific transcriptome libraries were built, and sequencing was performed using the Illumina NovaSeq6000 system, which generates stranded paired-end sequences.

### 2.8. Bioinformatics

The RNA-Seq data analysis was performed in Chipster software [56]. Phred score was >35 along all reads in every sample, so no trimming was needed before read mapping. Reads were mapped to *P. cactorum* reference genome (GCA_010194725.1) with HISAT2. Reads were counted using HTSeq, and differential expression calls were made with edgeR and DESeq2 in Chipster v4 software. The raw reads are available in GenBank under BioProject PRJNA804427.

### 2.9. Proteomics

For proteome analysis, isolates were grown in liquid cultures (autoclaved mixture of 100 mL/L of vegetable juice premixed with 2 g/L of CaCO3 and 900 mL/L distilled H_2_0) for nine days, thoroughly washed in Tris-buffered saline (TBS) (150 mM NaCl, 50 mM Tris-HCl, pH 7.6) to remove any residues of the cultivation medium, then frozen in liquid nitrogen and lyophilized. The proteome analysis was performed with liquid chromatography-mass spectrometry as previously described [57]. In brief, lyophilized samples were extracted with tert-butyl methyl ether/methanol mixture, proteins were solubilized, digested with trypsin, and analyzed by nanoflow reverse-phase LC-MS using a 15 cm C18 Zorbax column (Agilent, Santa Clara, CA, USA), a Dionex Ultimate 3000 RSLC nano-UPLC system, and the Orbitrap Fusion Lumos Tribrid Mass Spectrometer (Thermo Fisher, Waltham, MA, USA). The measured spectra were recalibrated and compared against the databases of *P. cactorum* [58] and common contaminants using Proteome Discoverer 2.5 (Thermo Fisher) with Sequest HT and MS Amanda 2.0 [59] algorithms. The quantitative differences were determined by Minora, employing precursor ion quantification followed by normalization and calculation of relative peptide/protein abundances. The results were evaluated by a background-based t-test, and the resulting p-values were adjusted using the Benjamini-Hochberg method. Only proteins with at least two unique peptides were considered for the quantitative analysis. The mass spectrometry proteomics data were deposited at the ProteomeXchange Consortium via the PRIDE [60] partner repository with the data set identifier PXD033832.

### 2.10. Pathogenicity

In order to test for virulence of PhF66 and PhF66– to strawberry, two inoculation tests were set up in a greenhouse in October-December 2020 (strawberry test I). Runner plugs of greenhouse-maintained strawberry plants of varieties Glima, Jonsok, Senga Sengana, Zefyr, and Honeoye were rooted in rockwool grow cubes (2 × 2 × 3 cm; Grodan, Roermond, the Netherlands) two weeks prior to inoculation. The well-rooted plug plants were then randomized and spaced into larger rockwool plates fitted in plastic containers, which were subsequently kept on a table in a greenhouse bench under +23 °C and long day (19 h) conditions. Six plants of each variety per each inoculation treatment were wounded in the stem (crown) base with a sterile blade tip, and a malt agar plug containing mycelium of PhF66, PhF66– or sterile agar plugs containing no hyphae (mock-inoculated control plants) was placed on the wound. The inoculum was covered with moist cotton and sealed with parafilm. The cotton was removed after five days. The experiment was monitored for 30 days, and repeated twice, two weeks apart. Strawberry test II was set up in June 2021 with strawberry plants (Malling Centenary) planted in the previous summer and growing outdoors in plastic growth bags filled with peat (Kekkilä Growth Sack Airboost 1 m). The plants were inoculated on 28 May 2021, as described above, with an agar plug containing PhF66 (four plants), PhF66– (four plants), PhF17/19 (six plants) –another *P. cactorum* strain isolated from strawberry [10]– or no hyphae (mock-inoculated control; four plants). Leaves were counted and scored healthy/wilting/dead once a week for three weeks. Furthermore, one-year-old silver birch (*Betula pendula*) seedlings were inoculated with isolates PhF66 (15 seedlings), PhF66– (15 seedlings), Ph415 –another *P. cactorum* strain isolated from birch (Table 1; [61])–(13 seedlings), and agar-control (10 seedlings) in June 2021. Birch stem inoculations were performed similarly to that of strawberries, by wounding the bark and placing an malt agar plug containing hyphae onto the wound [62]. The longitudinal lengths of the resulting lesions were measured at 6, 13, and 20 days post infection. Differences in lesion sizes between isolates were analysed by one-way ANOVA in IBM SPSS Statistics.

## 3. Results

### 3.1. Curing Virus Infections

All isolates grew in all water matrix potentials applied in the experiment, but their growth was significantly reduced at −13 MPa and −15 MPa compared to growth rates at −11 MPa. The two replicates of each isolate (PhF9, PhF66, and PhF79) recovered from water potentials −13 MPa and −15 MPa were subjected to RNA extraction, reverse transcription, and virus-specific PCR. Amplification products of alphaendornaviruses PcAEV1, PcAEV2, and PcAEV3 were obtained from all replicates of the isolate hosting them (PhF9). The toti-like PcRV1 was also detected from all replicates of isolates PhF9 and PhF79. Phytophthora cactorum usti-like virus (PcUV1) as well as the bunyavirus PcBV2 were detected in all PhF79 replicates. However, the bunyavirus PcBV1 was not detected in PhF79, and both PcBV1 and PcBV2 were cured from all four analyzed replicates of PhF66. The result was confirmed by RNA-seq analysis of isolate PhF66–, where no reads of virus sequences were obtained.

### 3.2. Transfer Tests

Mycelial samples taken from the recipient (the virus-free PhF66–) side of the dual cultures with virus hosting strains (PhF38, PhF66, PhF79, and PhF101) were analyzed for the presence of transmitted viruses using RT-PCR with virus-specific primers [10]. Viruses PcEV1, PcAEV2, PcAEV3, PcBV1, PcBV2, PcBV3, PcRV1, and PcUV1 were screened according to the viruses in the donor, but no amplification was observed at 21 or 64 days. Therefore, it was concluded that no successful virus transmissions had occurred.

### 3.3. Viral Effects on Host Growth, Sporulation and Morphological Characteristics

To test the phenotypic effects of bunya-like viruses in laboratory conditions, we conducted growth experiments using strains PhF66 and PhF66–, hosting two bunyaviruses and no viruses, respectively. The presence of these viruses significantly lowered the growth rate of the host isolate PhF66 in both growth substrates. There was a significant effect of the virus status on the growth rate of PhF66. The radial growth rate of the bunyavirus-containing PhF66 on MEA was 2.3 mm/day (range 2.20–2.51 mm/day) and that of PhF66– was 3.1 mm/day (range 2.89–3.24 mm/day), which differed significantly from each other (F = 936, *p* < 0.001, N = 44).The mean lesion areas on apples were 10.0 cm^2^ (range 1.85–18.18 cm^2^) and 15.8 cm^2^ (range 1.97–31.14 cm^2^), respectively, which also differed significantly (U = 149, *p* < 0.05, N = 44)

To examine the effects of the two bunya-like viruses on sporangia formation and zoospore release, three experiments using different settings were conducted. Sporangia were produced by all three methods, but no reliable zoospore counts were obtained. In experiment 1, the average number of encysted zoospores in the 12 uL examined for each replicate in Bürker chamber was 2.66 for PhF66 and 4.2 for PhF66–, which was considered too low for a reliable count. In experiments 2 and 3, empty sporangia and a few motile zoospores (Figure 1a) were observed under a microscope, but later on no reliable counts of encysted zoospores were obtained. At the time of the expected zoospore release, some sporangia germinated directly by the formation of a germination tube instead of producing zoospores (Figure 1b). In addition, in many cases, there was no evidence of zoospore differentiation inside the sporangia, and cytoplasm would often be released out of them instead of biologically viable zoospores (Figure 1c). In experiment 2, the formation of young sporangia was recorded in both isolates and all replicates at 16 h after the first flooding.

Differences in sporangia production and morphological characteristics were observed between isolates PhF66 and PhF66–. PhF66 produced significantly less (*p* < 0.01) and smaller (*p* < 0.0001) sporangia compared to PhF66– (Table 2). No differences between the isolates were found in the pedicel size, oogonia size, oospore diameter, thickness of oospore walls, or antheridia size. Oospore abortion rates were relatively high in both variants but significantly higher (*p* < 0.001) in the virus-free PhF66– (Table 2).

### 3.4. Viral Effects on the Host Gene Expression and Protein Abundance

The overall alignment rate of the raw RNA-seq reads to the P. cactorum reference genome was 94.81–95.27% between the samples. While summarizing mapped reads into a gene level count, 51.0–54.4% of reads were not counted in the samples (mostly due to non-unique alignments and less than complete annotation of the reference genome). All three biological replicates of PhF66 differentiated significantly from those of PhF66– in the principal component analysis and heatmap (Appendix A). A batch effect was also found between the first biological replicate (six samples) and biological replicates 2 and 3 (four samples) (Appendix A), which was taken into account as an additional factor in the differential expression analysis. The number of differentially expressed genes (DEGs) having *p*-values < 0.05 with both egdeR and DESeq2 was 1824. Out of the 1824 DEGs, 120 had cutoff values greater than 1.5 (Appendix A), and in the case of 33 genes, the cutoff value was greater than 2 (representing over 4× fold change) (Table 3). Ten of the 33 DEGs represented hypothetical proteins, but a conserved domain was found in 23 DEGs. Overall, 10 genes were upregulated in the virus-containing isolate, three of which were elicitin genes (Table 3). Notably, no transcripts of the gene coding for hypothetical protein GQ600_18332 were obtained from the virus-free isolate PhF66–, but it was abundantly transcribed in PhF66 (Table 3). The amino acid sequence of protein GQ600_18332 had 43.8% identity with the DDE_Tnp_1_7 domain containing protein of Phytophthora palmivora in Uniprot BLAST. The domain is found in the PiggyBac transposable element-derived proteins (PGBDs) as well as in some uncharacterized proteins, but its function is not known [63]. Among the DEGs with lower cut-offs (1.5–2), upregulated genes also included, e.g., a CAP (cysteine-rich secretory proteins, antigen 5, and pathogenesis-related 1) domain that likely functions in sequestering sterols from plants [64,65], a C2H2 zinc finger domain that has putative roles in oospore development and virulence [66], and a “necrosis inducing protein” which refers to NPP1-like proteins in Oomycetes capable of inducing ROS and HR-like cell death in the host [67] (Appendix A).

Out of the 23 genes that were downregulated over 4× fold in the virus-containing isolate, four included a WD40 repeat-containing domain (Table 3). WD40 domain-containing proteins are found in all eukaryotes, and they function as coordinators of protein–protein interactions [68]. The ankyrin-repeat domain (found among up- and downregulated DEGs) has a similar function [69]. Among the DEGs with cutoffs 1.5–2, downregulated genes included, e.g., two additional WD40 repeat domains and three genes with Leucine-rich repeat (whose function is largely unknown in Oomycetes but reported to be involved in zoospore flagellum development in P. sojae [70]).

In the proteomic analysis, 1392 proteins were detected, out of which 53 differed significantly (*p* < 0.05) in their abundance between PhF66 and PhF66–. The differentially abundant proteins included, e.g., two elicitins and various enzymes (Table 4). When comparing the data from transcriptomic and proteomic analyses, only one of the 33 DEGs with cut-offs >2 were found among the 53 differentially abundant proteins–namely an elicitin. Two additional differentially expressed genes were identified among the detected non-differentally abundant proteins. When searching for the 53 differentially abundant proteins in the RNA-seq data, only six of them were found among the 1824 differentially expressed genes (*p* < 0.05) (Table 4).

### 3.5. Viral Effects on Host Pathogenicity

The virulence of isolates PhF66 and PhF66– to strawberry was tested in two occasions. In the strawberry test I which included five strawberry cultivars and two repetitions, no typical symptoms of crown rot appeared during the 30-day experiment. Cross-sections of crown stems of the strawberry plants were inspected, and re-isolation of the pathogen was attempted at 30 dpi. However, no lesions or browning were found in the internal tissues, and pathogen isolations were unsuccessful. In the strawberry test II, the first wilting of leaves was observed on two plants at 4 dpi, on plants inoculated with PhF17/19 (positive control), and the first dead plant was observed at 12 dpi. The last inventory was made at 20 dpi, when all plants inoculated with PhF17/19 were dying. No symptoms were found on plants inoculated with PhF66 or PhF66– (Figure 2a,b). Re-isolations were attempted at 20 dpi from all inoculations, but they were successful only from plants infected with PhF17/19 (four re-isolations). The results are shown in Table 5.

In the pathogenicity test with silver birch, lesions in all inoculated plants were observed at 6 dpi (Figure 2c). However, a very small increase in lesion length was measured at 13 dpi. No lesion growth was recorded after 13 dpi, indicating the subsequent healing of lesions most likely due to very high temperatures (over 30 °C) for several days at the time of the test, which may have slowed down the pathogen growth and allowed the seedlings to suppress its advancement in the stem tissues. Re-isolations were attempted at the time of last inventory at 20 dpi, but with no success. ANOVA was performed with 13 dpi measurements, which returned no significant differences in lesion sizes induced by the three isolates (F = 0.077, *p* = 0.926, N = 41, Figure 3). No lesion development was detected among the mock-inoculated control seedlings.

## 4. Discussion

Bunya-like viruses have been described from the oomycetes *P. condilina* [13], *P. castaneae* [20], *Halophytopththora* spp. [47], *Globisporangium polare* [46], as well as many fungal hosts, but to our knowledge there are no previous reports on the direct effects of bunyavirus infections on fungal or oomycete hosts. In the present study, infection by two bunyaviruses (PcBV1 & PcBV2) reduced the growth rate of the host isolate (PhF66) on MEA plates as well as in apple tissue. Two endornaviruses have also been shown to inhibit mycelial growth in *Phytophthora* sp. from asparagus, but in contrast to the results of the current study, sporangia production was enhanced [11].

Both transcriptomic and proteomic analyses indicated that certain elicitins were differentially expressed in the virus-infected and virus-free isolate. The expressions of three elicitins were strongly upregulated at the RNA level, and one of them was also more abundant at the protein level in the virus-containing isolate (Table 3 and Table 4). Elicitins are structurally conserved proteins excreted by Oomycetes, mainly by species of *Phytophthora* and *Pythium*, during host plant interaction. They are apoplastic effectors that are utilized to suppress induced immunity responses in the host plant [71,72,73]. While elicitins promote oomycete virulence, in some pathosystems, they can in turn become recognized by the host defense system and become associated with reduced pathogenicity [74,75]. The translation initiation of a gene putatively coding for a PiggyBac transposable element-derived protein in the virus-containing isolate may indicate viral effect on transposon activity in the host, although no proteins including this domain were found among the proteins differing in their abundance. The downregulated DEGs in the virus-containing isolate included many genes that function as mediators of protein–protein interactions in multi-protein complexes (WD40 and ankyrin repeat domains). While WD40 domain-containing proteins have been identified as negative regulators of *Arabidopsis* resistance to *Phytophthora* [76,77], it is also found in, e.g., cyclophilin-encoding genes in the *Phytophthora* species, which potentially have various roles in plant infection [78]. The WD40 domain is also found in the avirulence genes of *P. infestans*, *P. sojae*, and *P. ramorum,* coding for cytoplasmic effectors that trigger an immune response in the host [79,80,81].

Relatively few of the DEGs were found to be differentially abundant as proteins and vice versa. While RNA sequencing enables the profiling of slight changes in the transcript expression between two conditions, it does not capture the involvement of various post-transcriptional, post-translational, and degradation pathways that influence the relative quantities of expressed protein [82]. Thus, RNA transcript and protein abundance are often only weakly correlated [83]. Furthermore, the mRNAs (differentially) expressed at low levels may be below the detection limit at the protein level. It should be noted that the isolates were grown on V8 growth media instead of MOS prior to the proteomic analyses, which likely resulted in some differences in the gene and protein expressions.

In the pathogenicity tests conducted with the bunyavirus-containing and virus-free isolates (PhF66 and PhF66–), both isolates failed to cause symptoms in strawberry. However, lesions were induced on silver birch indicating the isolate may belong to the population of *P. cactorum* preferring birch as a host. However, we did not observe any differences between the virus-free and virus-containing isolates in pathogenicity to birch. Thus, understanding the roles and functions of the elicitins upregulated by these bunyaviruses requires further investigation.

The PhF66 isolate containing the bunyaviruses also produced less and smaller sporangia. However, no effective zoospore release was documented. Stimulation of sporangia production was also reported in *P. infestans* by the unclassified virus PiRV-2 [33]. The present study also found relatively high oospore abortion rates in both isolates (PhF66 & PhF66–), which was significantly higher in the virus-free isolate. In an evolutionary analysis of European *P. cactorum* populations, Pánek et al. [84] inferred Finnish *P. cactorum* isolates (representing woody species pathotype) to possibly be of hybrid origin. The isolates were also characterized by high oogonial abortion rates, suggesting that the putative hybrid status could be the reason for the relatively high oospore abortion rates detected in the present study. The higher oospore abortion rate in the virus-free isolate may also be related to stress caused by the virus-curing treatment in low water potential. Adverse conditions such as the long deep-freeze storage or previous exposure to fungicides in the field can decreaseoospore viability (T. Jung, pers. comm.).

In this study, we found that the most prevalent virus infections (endorna- and toti-like viruses) in *P. cactorum* are highly stable and resist curing attempts by osmotic stress, but the rarer infections of bunyaviruses seem less stable and more easily curable. In general, the curing and transmission of viruses in vitro has been challenging with the *Phytophthora* species [33,85]. In the present study, to avoid mutations in the host caused by chemical treatments such as Ribavirin, we tested the suitability of PEG-induced stress on water availability for curing viruses from *Phytophthora*. We found that the *P. cactorum* isolates were very persistent during low water availability conditions and were able to grow in and were recoverable from all tested water potentials, down to −15 MPa. The method was originally developed and proved successful for the fungus *Pseudogymnoascus destructans*, whose growth ceased at −5 MPa, and an inhabiting partitivirus was cured already at −2 MPa [51]. In our study, the more nutrient-rich medium (potato dextrose) could have favored the growth of *Phytophthora* isolates and thus explain its viability in the very low water potentials during this study. All three alphaendornaviruses (+ssRNA; PcAEV1-3), a toti-like virus (dsRNA; PcRV1), and an ustivirus (dsRNA; PcUV1) remained in the host isolates during the treatment. However, bunyaviruses (-ssRNA; PcBV1, PcBV2) were shown to be curable with this method. Previously, the unclassified dsRNA virus Phytophthora infestans RNA virus 2 (PiRV-2) and a member of “Fusagraviridae”, Phytophthora infestans RNA virus 3 (PiRV-3), were cured successfully after four and five hyphal tip transfers on Ribavirin-containing plates, respectively [16,33]. Curing was reported unsuccessful for a narnavirus (PiRV4, [15]) and the unclassified Phytophthora infestans RNA virus 1 (PiRV1, [85]). In the present study, the duration of the treatment was limited to three weeks. The usefulness of additional combinations of water potentials, lower nutrient media, and longer test duration for curing additional viruses from *Phytophthora* may be subjects of further study. In conclusion, PEG-induced water stress is a potential method for curing *Phytophthora* viruses, but efficacy varies between viral species.

Despite viruses being highly common in isolates of *P. cactorum* [10], we found that they are not readily transmissible by hyphal contacts in laboratory conditions, as the virus transfer between *P. cactorum* isolates in dual cultures was not successful. Virus transmission by hyphal anastomosis has been attempted but also proved relatively challenging for P. *infestans* [85]. However, Cai et al. [33] reported a successful reintroduction of PiRV-2 into the same isolate after two weeks on joint rye agar plates. In the present study, the dual cultures of different strains were maintained for 64 days on MEA, and no virus transfer was observed even between isogenic PhF66 and PhF66–. Overall, it seems that the viral transfer via mycelial contact is not a frequent event, even between isogenic *Phytophthora* isolates in laboratory conditions. At a population level, *P. cactorum* was recently shown to host multiple viruses with an extremely high prevalence (99% of isolates hosted viruses), indicating a long co-evolution likely resulting in neutral or positive symbiotic relationships [10]. Furthermore, the pathotypes infecting strawberry seem to share the same viruses globally, suggesting the transport of *P. cactorum* along with host plants between continents [10]. However, the presence of the same viruses in multiple host isolates can also result from a higher vertical transmission potential between host genotypes in the field compared to laboratory conditions, and it should also be noted that species of *Phytophthora* are prone to hybridization [86], which could provide an additional route for virus transmission.

In summary, the present study showed that bunyaviruses, but not other tested viruses, may be cured from *P. cactorum* by PEG-induced water stress, although transmission between different mycelia in laboratory conditions seems to be inefficient. Bunyaviruses were shown to cause phenotypic effects on the growth rate of *P. cactorum* on artificial media and in apple fruits, but not on the pathogenicity. Bunyaviruses also induced the upregulation of elicitin genes, one of them at both transcript and protein levels, in the virus-containing isolate. Upregulation of other apoplastic effectors (CAP-family, necrosis-induced proteins) was also observed at the transcriptional level, indicating a putative viral effect on host pathogenesis.

## Figures and Tables

**Figure 1 viruses-14-02596-f001:**
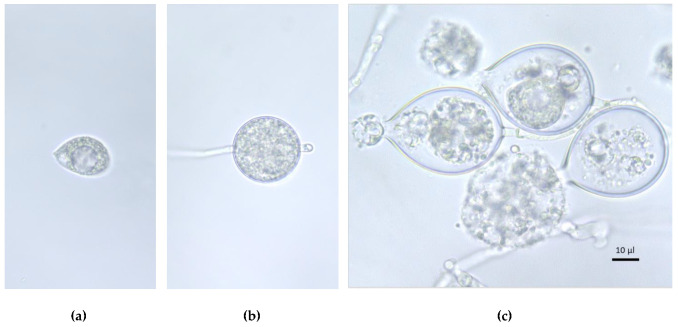
Morphological features of zoospores and sporangia, and sporulation pattern of *P. cactorum* isolate PhF66– on clarified VA. (**a**) A zoospore observed after stimulated zoospore release; (**b**) direct proliferation of PhF66– sporangium, and (**c**) release of cytoplasm from the sporangia. Scale bar = 10 µm.

**Figure 2 viruses-14-02596-f002:**
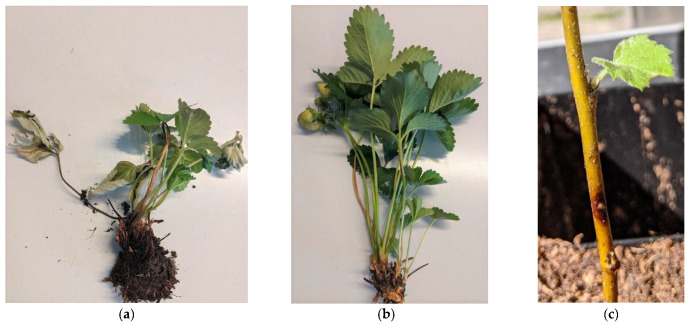
*Fragaria* × *ananassa* (Malling Centenary) 20 days post inoculation with *Phytophthora cactorum* isolates PhF17/19 (**a**) and PhF66 (**b**). (**c**) *Betula pendula* seedling six days post inoculation with *P. cactorum* isolate PhF66.

**Figure 3 viruses-14-02596-f003:**
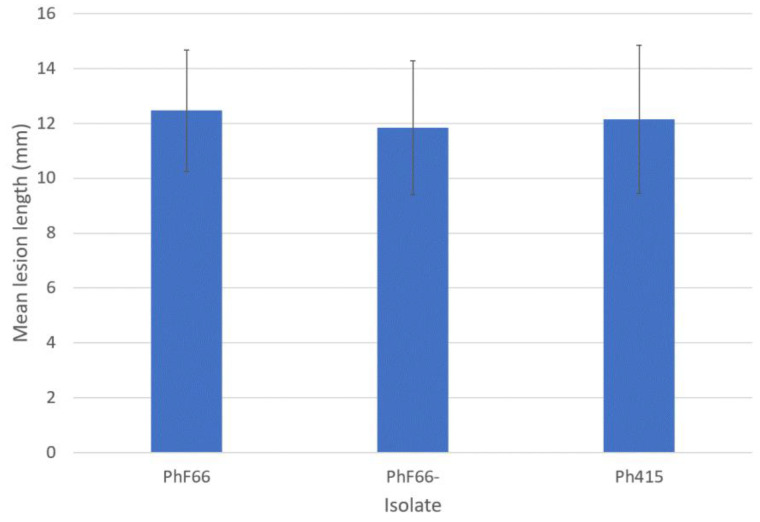
Mean lesion lengths from *Phytophthora cactorum* pathogenicity test on silver birch at 13 dpi. Error bars: 95% CI.

**Table 1 viruses-14-02596-t001:** *Phytophthora cactorum* isolates used in this study. The viruses infecting these isolates are also given. PcAEV1–3 = Phytophthora cactorum alphaendornavirus 1–3; PcBV1–3 = Phytophthora cactorum bunya-like viruses 1–3; PcRV1 = Phytophthora cactorum RNA virus 1 (putative *Totiviridae*; [14]); PcUV1 = Phytophthora cactorum usti-like virus (PcUV1). The isolates were used in the following analyses of this study; I = virus curing, II = virus transmission, III = virus effects (growth, sporulation, pathogenicity, transcriptome, proteome), IV = pathogenicity.

Isolate Name	Viruses	Isolation Year	Strawberry Cultivar	Collection Location	Used in Analyses
PhF9	PcAEV1, PcAEV2, PcRV1	1990	Jonsok	Pohja	I
PhF38	PcAEV1, PcAEV2, PcRV1	1993	unknown	Ilomantsi	II
PhF66	PcBV1, PcBV2	2006	Polka	Nurmijärvi	I, II, III, IV
PhF79	PcBV1, PcBV2, PcBV3, PcRV1, PcUV1	1991	Jonsok	Pohja	I, II
PhF101	PcAEV1, PcAEV3, PcRV1	1993	Jonsok	Pohja	II
PhF17/19	PcRV1	2019	Polka	Huhdasjärvi	IV
Ph415	unknown	2004	*Betula pendula*	Suonenjoki	IV

**Table 2 viruses-14-02596-t002:** Morphological characteristics measured from the virus-free *P. cactorum* isolate PhF66– and PhF66 containing two bunyaviruses.

		Sporangia				Oogonia					Antheridia			
		Length/µm	Breadth/µm	Sporangia Form	Abundance (№ per Agar Plug)	Length /µm	Breadth /µm	Oogonia Form	Oospore Diam./µm	Oospore Wall/µm	Length /µm	Breadth /µm	Antheridia Form	Abortion Rate
PhF66	Mean Min Max	36.37 26.35 45.11	29.77 22.14 37.05	broad-ovoid and ovoid, also spherical	897 453 599	32.49 27.19 39.09	30.60 24.16 35.24	smooth, globous to sub-globous	25.87 21.57 30.79	1.27 0.44 1.83	14.52 9.55 26.41	10.50 7.01 23.52	Paragynus, clavate	29.27% 20% 38%
PhF66–	Mean Min Max	45.10 33.87 57.13	35.58 29.72 42.75	ovoid to slight limoniform	519 724 1148	33.66 28.24 39.12	29.81 24.57 33.64	smooth, globous to sub-globous	25.81 21.57 32.81	1.33 0.84 1.84	14.04 9.53 22.88	9.98 5.86 12.51	Paragynus, clavate	91.11% 85% 97%
N		50	50	50	2076/3588 (PhF66/PhF66–)	50	50	50	50	50	50	50	50	100
*p*-value sig. level ≤0.00625		4.207 × 10^−13^	1.135 × 10^−12^		0.008073	0.03295	0.06979		0.8706	0.2227	0.4411	0.2362		0.000586

**Table 3 viruses-14-02596-t003:** Differentially expressed genes in the virus free *P. cactorum* isolate PhF66– and PhF66 containing two bunyaviruses with cutoff values >2 revealed by RNA-seq. * = corresponding proteins detected differentially abundant alsoin the shotgun proteomic analysis. ** = corresponding proteins detected, but not showing differences in abundance. Annotated elicitins are highlighted in bold.

			Read Counts per Million												
			PhF66– (No Viruses)					PhF66 (PcBV1&PcBV2)							
Accession	Description	Length	R1	R2	R3	R7	R8	R4	R5	R6	R9	R10	log2Fold Change	FDR	Padj
KAF1787014.1	hypothetical protein GQ600_18332	581	0.00	0.00	0.00	0.00	0.00	12.91	17.78	21.27	21.19	15.59	6.56	0.000002	<0.001
KAF1778147.1	Riboflavin synthase-like beta-barrel	1444	1.50	0.92	1.54	4.68	1.66	22.08	54.90	8.28	8.28	10.26	2.70	0.000053	<0.001
**KAF1774323.1**	**Elicitin**	**534**	**52.63**	**47.31**	**47.61**	**7.90**	**8.57**	**45.43**	**77.86**	**283.71**	**293.98**	**292.02**	**2.68**	0	**<0.001**
**KAF1774322.1**	**Elicitin**	**516**	**0.11**	**0.16**	**0.00**	**0.08**	**0.00**	**0.07**	**0.25**	**2.92**	**2.77**	**2.32**	**2.49**	0.000513	**<0.001**
KAF1789104.1	hypothetical protein GQ600_9925	270	0.75	0.84	0.35	0.45	0.58	4.82	5.01	4.02	2.77	3.87	2.43	0.00005	<0.001
KAF1774139.1	hypothetical protein GQ600_10043	470	0.15	0.08	0.04	0.12	0.12	0.91	2.96	0.69	1.26	1.22	2.39	0.000147	<0.001
**KAF1781019.1 ***	**Elicitin**	**357**	**1.94**	**0.72**	**1.41**	**0.04**	**0.00**	**0.98**	**1.34**	**6.99**	**28.63**	**13.27**	**2.35**	0.000294	**<0.001**
KAF1780261.1	SMP-30/Gluconolactonase/LRE-like region	1114	2.13	2.32	1.19	2.65	1.12	15.59	14.53	8.43	7.82	7.16	2.19	0.000161	<0.001
KAF1780813.1	hypothetical protein GQ600_560	264	1.31	1.64	0.84	0.90	0.42	7.40	2.34	7.39	5.93	6.80	2.14	0.000655	<0.001
KAF1785474.1	Nucleotide-diphospho-sugar transferase	879	0.82	0.32	0.66	2.28	0.87	6.96	19.45	2.43	2.52	2.77	2.07	0.000655	<0.001
KAF1788825.1	Ankyrin repeat-containing domain	1174	34.09	38.83	36.01	33.31	39.71	0.00	0.00	0.25	0.29	0.24	−3.70	0	<0.001
KAF1791464.1	WD40/YVTN repeat-like-containing domain	5799	5.27	4.96	5.47	4.56	5.41	0.36	0.04	0.30	0.50	0.45	−3.24	0.000041	<0.001
KAF1788826.1	hypothetical protein GQ600_3050	762	2.73	1.96	1.94	0.49	1.70	0.00	0.00	0.00	0.00	0.00	−3.21	0.000421	<0.001
KAF1775491.1	WD40-repeat-containing domain	6238	2.77	2.20	2.60	2.52	1.58	0.11	0.38	0.10	0.04	0.08	−2.72	0.000065	<0.001
KAF1791150.1	hypothetical protein GQ600_24668	3162	9.79	11.80	12.58	13.97	13.02	3.08	1.54	0.69	0.76	0.85	−2.69	0.000062	<0.001
KAF1789612.1	putative domain, di-copper centre	1712	15.59	12.76	13.94	5.33	4.95	0.98	0.38	2.28	1.89	1.83	−2.54	0.000029	<0.001
KAF1790519.1	putative domain, di-copper centre	444	2.80	2.64	3.35	1.18	0.75	0.22	0.00	0.30	0.13	0.28	−2.45	0.000347	<0.001
KAF1789787.1	hypothetical protein GQ600_24324	311	4.45	3.72	3.79	2.00	1.04	0.25	0.21	0.50	0.63	0.16	−2.34	0.000161	<0.001
KAF1792605.1	hypothetical protein GQ600_23700	6403	11.51	13.08	11.21	29.07	23.20	4.50	3.42	2.33	2.40	1.91	−2.32	0.000008	<0.001
KAF1795150.1	Tetratricopeptide repeat	7222	16.15	15.16	16.02	30.86	24.70	4.68	4.97	3.57	2.48	2.61	−2.31	0.000001	<0.001
KAF1782989.1	WD repeat-containing protein 90	7515	5.05	4.92	5.47	5.13	4.32	1.20	1.13	0.45	0.67	0.28	−2.29	0.000164	<0.001
KAF1782797.1 **	Thrombospondin type-1 (TSP1) repeat	7184	67.62	56.07	58.42	14.66	11.35	1.99	2.46	12.74	11.94	9.53	−2.29	0.000001	<0.001
KAF1783174.1	hypothetical protein GQ600_25520	558	0.78	0.84	0.88	0.41	0.71	0.00	0.00	0.00	0.04	0.04	−2.28	0.000328	<0.001
KAF1775375.1 **	Carbonic anhydrase, alpha-class, conserved site	661	24.45	20.76	23.17	1.55	1.75	0.51	0.75	1.74	2.77	1.83	−2.24	0.000232	<0.001
KAF1793614.1	hypothetical protein GQ600_19168	3971	9.38	9.60	9.58	10.42	13.06	1.81	1.34	2.58	1.85	1.55	−2.23	0.00003	<0.001
KAF1783504.1	WD40-repeat-containing domain	3039	1.16	2.12	1.72	1.59	2.49	0.33	0.38	0.20	0.13	0.00	−2.21	0.000635	<0.001
KAF1783853.1	Katanin p60 subunit A1	2091	11.85	15.64	14.21	12.54	14.97	3.44	3.42	1.98	1.64	2.48	−2.19	0.000019	<0.001
KAF1793060.1	EF-Hand 1, calcium-binding site	804	1.46	1.92	1.85	1.26	0.62	0.04	0.13	0.10	0.17	0.33	−2.18	0.000247	<0.001
KAF1772692.1	EF-hand domain pair	2739	1.31	1.20	1.94	1.34	0.83	0.15	0.13	0.00	0.00	0.00	−2.12	0.000757	<0.001
KAF1789078.1	WD40-repeat-containing domain	2961	2.80	3.76	3.22	3.91	4.12	0.80	1.04	0.25	0.38	0.69	−2.12	0.000098	<0.001
KAF1794169.1	Organic solute carrier protein 1	5303	5.53	6.96	6.35	11.81	10.35	3.37	3.46	0.79	0.42	0.28	−2.11	0.000655	<0.001
KAF1782147.1	P-loop containing nucleoside triphosphate hydrolase	13632	6.65	5.24	4.85	7.94	8.48	1.34	2.00	1.69	0.63	0.69	−2.10	0.000126	<0.001
KAF1788166.1	hypothetical protein GQ600_1662	1674	0.52	0.60	0.26	0.90	0.50	0.00	0.00	0.00	0.00	0.00	−2.04	0.002536	<0.001

**Table 4 viruses-14-02596-t004:** Differentially abundant proteins (*p* < 0.05) in the virus free PhF66– (samples S2–4) and PhF66 (samples S5–7) containing two bunyaviruses by RNA-seq. * = proteins also found differentially expressed in RNA-seq (*p* < 0.05). “High” and “Peak Found” represent proteins identified by matching peptide MS/MS spectra and proteins that were identified by matching combinations of expected peptide mass and retention time, respectively. Annotated elicitins are highlighted in bold.

				PhF66– (No Viruses)			PhF66 (PcBV1&PcBV2)				
Accession	Description	Unique Peptides	Blank	S2	S3	S4	S5	S6	S7	log2Fold Change	Padj
KAF1782669.1	S-adenosyl-L-methionine-dependent methyltransferase	2	Not Found	Not Found	Not Found	Not Found	High	High	Peak Found	6.64	<0.001
KAF1784643.1	P-loop containing nucleoside triphosphate hydrolase	2	Peak Found	Not Found	Not Found	Not Found	Not Found	High	Peak Found	6.64	<0.001
KAF1791085.1	hypothetical protein GQ600_7093	5	Not Found	Not Found	Peak Found	Peak Found	High	High	High	4.69	<0.001
KAF1785983.1	FAD/NAD(P)-binding domain	2	Not Found	High	Not Found	High	Not Found	Not Found	Not Found	−6.64	<0.001
KAF1784304.1	Ubiquitin-activating enzyme E1, Cys active site	2	Not Found	Not Found	Not Found	Not Found	Not Found	High	Not Found	6.64	<0.001
KAF1777841.1	Zinc finger, RING/FYVE/PHD-type	2	Not Found	Not Found	Not Found	Not Found	Not Found	High	Not Found	6.64	<0.001
KAF1780096.1	50S ribosomal protein L30e-like	2	Not Found	Not Found	Not Found	Not Found	Not Found	High	Not Found	6.64	<0.001
KAF1778206.1	Nucleotide-binding alpha-beta plait domain	2	Not Found	Not Found	Not Found	Not Found	Not Found	High	Not Found	6.64	<0.001
KAF1772484.1	Exosome complex RNA-binding protein 1/RRP40/RRP4	2	Peak Found	Not Found	Not Found	Not Found	Not Found	High	Not Found	6.64	<0.001
KAF1791972.1 *	Zinc finger, NHR/GATA-type	2	Not Found	High	High	High	Not Found	Not Found	Not Found	−6.64	<0.001
KAF1794225.1	thiamine diphosphate-binding fold	2	Not Found	Not Found	Not Found	Not Found	High	High	Not Found	6.64	<0.001
KAF1775607.1 *	DNA polymerase family X lyase domain	2	Not Found	Not Found	Not Found	Not Found	Not Found	High	Not Found	6.64	<0.001
KAF1772539.1	Alpha/Beta hydrolase fold	2	Not Found	Not Found	Not Found	Not Found	Not Found	High	Not Found	6.64	<0.001
KAF1774000.1	hypothetical protein GQ600_3647	2	Not Found	Not Found	Not Found	Not Found	High	High	Not Found	6.64	<0.001
KAF1793066.1	Aldolase-type TIM barrel	2	Not Found	Peak Found	Not Found	High	Not Found	Not Found	Not Found	−6.64	<0.001
KAF1795828.1	Late embryogenesis abundant protein LEA	4	Not Found	Peak Found	High	Not Found	High	High	High	3.08	<0.001
KAF1790797.1	hypothetical protein GQ600_24592	4	Not Found	Peak Found	High	Peak Found	High	High	High	3.06	<0.001
KAF1781808.1	Leucine-rich repeat domain, L domain-like	3	Not Found	Not Found	Not Found	Peak Found	Not Found	High	Not Found	2.51	<0.001
KAF1790245.1 *	Fatty acid desaturase domain	2	Not Found	Not Found	Not Found	High	Not Found	Peak Found	Peak Found	−2.34	<0.001
KAF1772416.1	S-adenosyl-L-methionine-dependent methyltransferase	2	Not Found	High	High	High	Peak Found	Peak Found	Peak Found	−2.15	<0.001
**KAF1781019.1 ***	**Elicitin**	3	**Not Found**	**Peak Found**	**Not Found**	**Not Found**	**High**	**High**	**High**	2.30	<0.001
KAF1780998.1	Molybdenum cofactor biosynthesis, conserved site	2	Not Found	Peak Found	Not Found	Peak Found	Not Found	High	Not Found	2.22	<0.001
KAF1779922.1	hypothetical protein GQ600_6917	2	Not Found	Not Found	Peak Found	Not Found	High	High	Peak Found	2.08	<0.001
KAF1779895.1	Multicopper oxidase, type 3	2	Not Found	Not Found	Peak Found	Not Found	High	High	High	2.03	<0.001
KAF1794156.1	NAD(P)-binding domain	2	Not Found	Peak Found	Not Found	Peak Found	Not Found	High	Not Found	2.00	<0.001
KAF1783208.1	FAD/NAD(P)-binding domain	11	Peak Found	High	High	High	Peak Found	Not Found	Peak Found	−1.83	<0.001
KAF1789511.1	TmcA/NAT10/Kre33	2	Not Found	Not Found	High	Peak Found	High	High	High	1.91	<0.001
KAF1779266.1 *	Major facilitator superfamily domain	2	Not Found	Peak Found	Peak Found	High	Not Found	High	Peak Found	−1.73	<0.001
KAF1794053.1	NAD(P)-binding domain	2	Not Found	Peak Found	Peak Found	Not Found	High	High	Peak Found	1.88	<0.001
KAF1791322.1	Short-chain dehydrogenase/reductase, conserved site	10	Not Found	High	High	High	High	High	High	−1.72	0.001
KAF1787880.1	Multicopper oxidase, type 3	3	Not Found	Not Found	Peak Found	Not Found	High	High	High	1.84	0.002
KAF1780390.1	Ubiquitin conjugation factor E4, core	2	Peak Found	Peak Found	Not Found	Peak Found	Peak Found	High	Peak Found	1.83	0.002
KAF1779223.1 *	Glycoside hydrolase superfamily	6	Peak Found	High	High	High	High	High	High	1.83	0.002
**KAF1777766.1**	**Elicitin**	6	**Not Found**	**Peak Found**	**High**	**Peak Found**	**High**	**High**	**High**	1.80	0.002
KAF1773149.1	Thioredoxin-like fold	5	Not Found	High	High	High	High	High	High	1.73	0.003
KAF1780973.1	Leucine-rich repeat domain, L domain-like	2	Not Found	Peak Found	Not Found	Peak Found	Not Found	High	Peak Found	1.71	0.004
KAF1793851.1	Ubiquitin-conjugating enzyme, active site	20	Peak Found	High	High	High	High	High	High	1.66	0.006
KAF1791020.1	NAD(P)-binding domain	11	Peak Found	High	High	High	High	High	High	−1.45	0.009
KAF1781134.1	P-loop containing nucleoside triphosphate hydrolase	2	Not Found	Peak Found	Peak Found	High	Peak Found	Not Found	Not Found	−1.42	0.011
KAF1791905.1	FAD/NAD(P)-binding domain	2	Not Found	High	Peak Found	High	Peak Found	Peak Found	Peak Found	−1.37	0.016
KAF1788119.1	S-adenosyl-L-methionine-dependent methyltransferase	2	Not Found	High	Peak Found	High	Peak Found	Peak Found	Peak Found	−1.36	0.018
KAF1791684.1	ABC-transporter extension domain	4	Not Found	High	High	High	Peak Found	Peak Found	Peak Found	−1.33	0.021
KAF1794255.1	Peptidase C1A	4	Not Found	Peak Found	High	Peak Found	High	High	High	1.48	0.022
KAF1773151.1	Acyl-CoA dehydrogenase/oxidase, N-terminal	13	Peak Found	High	High	Peak Found	High	High	High	1.48	0.023
KAF1786762.1	HAD-like domain	2	Not Found	Not Found	Peak Found	Not Found	High	High	High	1.46	0.026
KAF1794937.1	Phosphoenolpyruvate carboxykinase (ATP), conserved site	3	Peak Found	High	High	High	Peak Found	High	High	−1.27	0.032
KAF1773405.1	Extradiol ring-cleavage dioxygenase, class III enzyme, subunit B	3	Not Found	High	Not Found	High	Not Found	Not Found	Peak Found	−1.24	0.037
KAF1773588.1	Glyoxalase/Bleomycin resistance protein/Dihydroxybiphenyl dioxygenase	2	Not Found	Peak Found	Not Found	Not Found	Peak Found	High	Peak Found	1.40	0.038
KAF1782519.1	ClpP/crotonase-like domain	14	Peak Found	High	High	High	High	High	High	1.38	0.043
KAF1775431.1	Cell division protein FtsZ, C-terminal	3	Not Found	Peak Found	Peak Found	Peak Found	Peak Found	High	Peak Found	1.37	0.046
KAF1778058.1	Protein argonaute, N-terminal	10	Peak Found	Peak Found	Peak Found	Peak Found	High	High	High	1.37	0.046
KAF1772871.1	Glycine cleavage T-protein/YgfZ, C-terminal domain	2	Not Found	High	High	High	Peak Found	Peak Found	High	−1.19	0.05

**Table 5 viruses-14-02596-t005:** Increments of healthy leaves in garden strawberry plants after inoculation with *Phytophthora cactorum* strains in the pathogenicity test II. dpi = days post inoculation.

Treatment	Replicate	0 dpi	20 dpi			Increase in Amount of Healthy Leaves	Average
		Healthy Leaves	Healthy Leaves	Wilting	Dead		
Control	1	5	8	0	0	3	
	2	3	6	0	0	3	
	3	4	8	0	0	4	
	4	3	6	0	0	3	**3.25**
PhF66	1	4	7	0	0	3	
	2	5	9	0	0	4	
	3	3	5	0	0	2	
	4	4	6	0	0	2	**2.75**
PhF66–	1	5	7	0	0	2	
	2	5	8	0	0	3	
	3	3	7	0	0	4	
	4	3	6	0	0	3	**3**
PhF17/19	1	5	7	1	2	−1	
	2	4	7	2	2	−1	
	3	5	7	5	2	−5	
	4	4	6	5	1	−4	
	5	4	7	0	4	−1	
	6	6	7	0	7	−6	**−3**

## Data Availability

The RNA-seq raw reads are available in GenBank under BioProject PRJNA804427, and the mass spectrometry proteomics data are available at the ProteomeXchange Central with the data set identifier PXD033832.

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
