# Peer review of "Bunyaviruses Affect Growth, Sporulation, and Elicitin Production in Phytophthora cactorum"

_viruses, 2022, doi:10.3390/v14122596_

Round 1
Reviewer 1 Report
Review of Bunyaviruses affect growth, sporulation and elicitin production 2 in Phytophthora cactorum
The manuscript of Poimala wt al provide a useful study on viruses of Phytophthora cactorum. This is a growing area of research in the last few years, and finding viruses and their functions in oomycetes has useful implications for virus biology, and for applied biocontrol.
The study is well written, and describes several well planned experiments. However, I feel some revisions (moderate-major) are needed before it is acceptable for publication in viruses. I hope the authors find my comments useful. I have provided general comments below, and also more specific comments in the annotated PDF.
One of my main concerns about the manuscript is that you have a lot of negative results. This is not a bad thing, as sometimes experiments do not yield the results we were expecting. However, what you need to be very careful about is not using your negative results to draw any strong conclusions. The most you should do with negative result is document them, explain why you think this might have happened. For example, there are 20 lines ( 519-539) on why the isolate did not infect strawberry. This could be cut to 2 sentences maybe.
Introduction
Your introduction is too long in my opinion. The authors are obviously well read in this area, but sometimes it feels to me that too much time is spent citing examples, when just a few examples would suffice.
Methods
You do not give any details of your statistical tests, software, which test was carried out on which experiment.
Results
Your results of your statistical tests should contain more than just the P value. It helps the reader to understand the extend of your data if you give number of replicates tested (n=X) . for example the text I found on the internet for reporting ANOVA indicates that this is a good way to report ANOVA: “There was a significant effect of amount of sugar on words remembered at the p<.05 level for the three conditions [F(2, 12) = 4.94, p = 0.027].” can you reuse this template for all of your reporting on statistical significance. You should provide information like this, to help the reader contextualise your results.
It’s a shame your pathogenicity tests didn’t reveal much for strawberry. You need to review your manuscript very carefully, and remove any conclusions about pathogenicity in strawberry.
Something strange happened to table 5 in the PDF proof I received, where it appears that all the table headings are repeated. Make sure this does not happen in subsequent drafts.
Many of your figures are unnecessary, and can be removed and results simply given in the text.
Discussion
You discussion is too long, and I have made some comments to help you reduce it. Stick to the main highlights of your work.

Author Response
We thank the reviewer for excellent comments to improve the manuscript. The answers to the general comments are given below, and answers to some of the the more specific comments are given in the additional pdf-file.
The manuscript of Poimala wt al provide a useful study on viruses of Phytophthora cactorum. This is a growing area of research in the last few years, and finding viruses and their functions in oomycetes has useful implications for virus biology, and for applied biocontrol.
The study is well written, and describes several well planned experiments. However, I feel some revisions (moderate-major) are needed before it is acceptable for publication in viruses. I hope the authors find my comments useful. I have provided general comments below, and also more specific comments in the annotated PDF.
One of my main concerns about the manuscript is that you have a lot of negative results. This is not a bad thing, as sometimes experiments do not yield the results we were expecting. However, what you need to be very careful about is not using your negative results to draw any strong conclusions. The most you should do with negative result is document them, explain why you think this might have happened. For example, there are 20 lines ( 519-539) on why the isolate did not infect strawberry. This could be cut to 2 sentences maybe.
ANSWER: This is true, we thank the reviewer for pointing this out. The text has been altered to better emphasize main results and better address the negative results.
Introduction
Your introduction is too long in my opinion. The authors are obviously well read in this area, but sometimes it feels to me that too much time is spent citing examples, when just a few examples would suffice.
ANSWER: The introduction has been shortened as suggested.
Methods
You do not give any details of your statistical tests, software, which test was carried out on which experiment.
ANSWER: We thank the reviewer for pointing out this missing information. Details on the statistical tests are now added to material and methods, and addressed in the text as suggested by the comments in the pdf.
Results
Your results of your statistical tests should contain more than just the P value. It helps the reader to understand the extend of your data if you give number of replicates tested (n=X) . for example the text I found on the internet for reporting ANOVA indicates that this is a good way to report ANOVA: “There was a significant effect of amount of sugar on words remembered at the p<.05 level for the three conditions [F(2, 12) = 4.94, p = 0.027].” can you reuse this template for all of your reporting on statistical significance. You should provide information like this, to help the reader contextualise your results.
ANSWER: The test statistic values and max-min ranges have been added to all statistical results and statements have been modified for better contextualization.
It’s a shame your pathogenicity tests didn’t reveal much for strawberry. You need to review your manuscript very carefully, and remove any conclusions about pathogenicity in strawberry.
ANSWER: As mentioned before, any conclusions regarding putative viral effects on the pathogenicity to strawberry have been removed.
Something strange happened to table 5 in the PDF proof I received, where it appears that all the table headings are repeated. Make sure this does not happen in subsequent drafts.
ANSWER: One row of data was accidentally left out of view; the table is now corrected.
Many of your figures are unnecessary, and can be removed and results simply given in the text.
ANSWER: Two figures were removed as suggested.
Discussion
You discussion is too long, and I have made some comments to help you reduce it. Stick to the main highlights of your work.
ANSWER: We thank the reviewer for excellent suggestions; the discussion has now been shortened and reorganized to better highlight main results.

Reviewer 2 Report
This manuscript showed impact of two bunyavirus infections on Phytophthora cactorum. The authors generated virus-free isogenic isolate and analyzed the effects of two bunyaviruses on mycelial growth, sporulation, virulence, and gene expression profile, however, I felt that this manuscript is descriptive. Unfortunately, I could not find scientific significance in this study. I recommend the authors should reconsider novelty and significance in this study and mention them properly based on the data.
Author Response
ANSWER: We have carefully gone through the manuscript and modified it according to all the comments received from all reviewers. The text has been modified to better highlight the main results to clarify and better emphasize the significance of the study. There is no previous knowledge on how bunyaviruses (or negative-sense RNA viruses in general) affect fungal or oomycete hosts, and the present study found effects on growth, gene & protein expression as we as sporangia production. The study also produced some negative results (pathogenicity and transmission potential). However, oomycete-virus systems are difficult to work on, so we hope the study will help expand the field of oomycete virology and provide additional information for researchers that may wish to work on such systems.
Reviewer 3 Report
Poimala et al used water stress to try to cure various viruses from Phytophthora cactorum isolates from strawberry and succeeded in curing two bunyaviruses from isolate PhF66. They compared isogenic isolates PhF66 and the cured PhF66- in growth rate, sporulation (sporangium, zoospore and sexual spores), pathogenicity, gene expression at RNA and protein level.
I have several comments about this manuscript:
1. I wish the authors had conducted virus transfer experiment between PhF66 and PhF66-. They did tried virus transfer between several other isolates and PhF66. It has been reported that virus transfer between different isolates of Phytophthora species through vegetative fusion was difficult but between isogenic isolates was possible. This may have something to do with vegetative incompatiblity system in Phytophthora which has not been well studied.
2. Given that isolate PhF66 was isolated from strawberry and their pathogenicity tests on strawberry was unsuccessful, I don't think they should make any conclusion about the virus impact on pathogenicity.
3. In Table 3, were the read counts raw counts or normalized counts (reads per million). The later should be used be used.
4. I wish the presentation was more concise and summarized. At present form, the authors described what happened in each trial, successful or otherwise.
Author Response
We thank the reviewer for excellent suggestions to improve the study.
Poimala et al used water stress to try to cure various viruses from Phytophthora cactorum isolates from strawberry and succeeded in curing two bunyaviruses from isolate PhF66. They compared isogenic isolates PhF66 and the cured PhF66- in growth rate, sporulation (sporangium, zoospore and sexual spores), pathogenicity, gene expression at RNA and protein level.
I have several comments about this manuscript:
- I wish the authors had conducted virus transfer experiment between PhF66 and PhF66-. They did tried virus transfer between several other isolates and PhF66. It has been reported that virus transfer between different isolates of Phytophthora species through vegetative fusion was difficult but between isogenic isolates was possible. This may have something to do with vegetative incompatiblity system in Phytophthora which has not been well studied.
ANSWER: The isogenic isolate PhF66 was indeed included in the transmission test, but it had dropped from the results (only considered shortly in discussion). Viruses were not indeed not detected even from the isogenic PhF66– after 64 days, despite good RNA quality and replicate isolates. PhF66 has been now added to the methods and results sections.
- Given that isolate PhF66 was isolated from strawberry and their pathogenicity tests on strawberry was unsuccessful, I don't think they should make any conclusion about the virus impact on pathogenicity.
ANSWER: Thank you, this is true. We have modified the manuscript to avoid making any conclusions on pathogenicity to strawberry.
- In Table 3, were the read counts raw counts or normalized counts (reads per million). The later should be used be used.
ANSWER: The table depicted actual number of reads counted per gene. The values have now been changed to normalized counts (reads per million).
- I wish the presentation was more concise and summarized. At present form, the authors described what happened in each trial, successful or otherwise.
ANSWER: The manuscript has been shortened and the structure slightly modified to better highlight main results.